# PROP1 triggers epithelial-mesenchymal transition-like process in pituitary stem cells

María Inés Pérez Millán*, Michelle L Brinkmeier, Amanda H Mortensen, Sally A Camper*

Department of Human Genetics, University of Michigan, Ann Arbor, United States

**Abstract** Mutations in *PROP1* are the most common cause of hypopituitarism in humans; therefore, unraveling its mechanism of action is highly relevant from a therapeutic perspective. Our current understanding of the role of PROP1 in the pituitary gland is limited to the repression and activation of the pituitary transcription factor genes *Hesx1* and *Pou1f1,* respectively. To elucidate the comprehensive PROP1-dependent gene regulatory network, we conducted genome-wide analysis of PROP1 DNA binding and effects on gene expression in mutant mice, mouse isolated stem cells and engineered mouse cell lines. We determined that PROP1 is essential for stimulating stem cells to undergo an epithelial to mesenchymal transition-like process necessary for cell migration and differentiation. Genomic profiling reveals that PROP1 binds to genes expressed in epithelial cells like C*laudin 23,* and to EMT inducer genes like *Zeb2, Notch2* and *Gli2. Zeb2* activation appears to be a key step in the EMT process. Our findings identify PROP1 as a central transcriptional component of pituitary stem cell differentiation.

*For correspondence: mipmillan@ gmail.com (MIPM); scamper@ umich.edu (SAC)

Competing interests: The authors declare that no competing interests exist.

## Introduction

Hereditary pituitary hormone deficiency, or hypopituitarism, is the most common pituitary disease in children and can cause significant morbidity if not treated effectively (*Vimpani et al., 1977*). There is a strong incentive for developing a pituitary stem cell therapy that could repopulate the pituitary hormone producing cells that are missing in patients with hypopituitarism. A comprehensive knowledge of pituitary stem cell differentiation is therefore relevant from a basic and medical research perspective.

Normal pituitary gland development involves invagination of oral ectoderm to create Rathke's pouch and evagination of the overlying neural ectoderm, which contains the FGF and BMP signaling center that stimulates pituitary growth. The stem cells in Rathke's pouch express SOX2, and as they transition to differentiation, they delaminate from the ventral aspect of the cleft, migrate ventrally and rostrally, and become the glandular parenchyma of the anterior lobe (*Ward et al., 2005*; *Rizzoti et al., 2013*; *Garcia-Lavandeira et al., 2009*). Notch signaling is required to prevent premature differentiation (*Zhu et al., 2006*; *Raetzman et al., 2004*), but the molecular mechanisms for stimulating progenitors to migrate and differentiate are unknown; and these processes are of central importance for understanding pituitary organogenesis and the pathophysiology of congenital pituitary hormone deficiency.

Mutations in several genes are known to cause hypopituitarism in children, and the pituitary transcription factors *HESX1, PROP1* and *POU1F1 (PIT1)* are among them. *PROP1* is the most commonly mutated gene, and it is the first pituitary-specific gene in the transcriptional hierarchy (*Agarwal et al., 2000*; *Deladoëy et al., 1999*; *Böttner et al., 2004*). PROP1 activity is modulated by WNT signaling, enabling it to suppress *Hesx1* and activate *Pou1f1* expression (*Olson et al.,*

*2006*). Despite its central role in pituitary organogenesis and important clinical significance, no comprehensive analysis of PROP1 function has been undertaken.

We hypothesized that PROP1 has a role in stem cell regulation because of the dysmorphic stem cell niche and cell migration defect in *Prop1* mutant mice, and because humans with *PROP1* mutations tend to have progressive hormone deficiency, which could be attributable to exhausting stem cell pools (*Böttner et al., 2004*; *Wu et al., 1998*). To test this idea, we used RNA-Seq and ChIP-Seq to identify novel targets and functions of PROP1. This led to the discovery that PROP1 has a key role in stimulating progenitors to undergo an epithelial-to-mesenchymal-like transition (EMT) prior to differentiation. PROP1 binds to promoters and enhancers of genes with demonstrated roles in EMT during development of other organs, including *Notch2, Zeb2* and *Gli2. Zeb2* expression appears to be a pivotal step in the EMT process. In addition, we show that PROP1 has an indirect role in regulating *Sox2* expression and stem cell proliferation. This in-depth molecular analysis of PROP1 action advances our fundamental understanding of pituitary organogenesis and the pathophysiology of hypopituitarism.

## Results

### PROP1 is transiently co-expressed with stem cell marker SOX2

PROP1 is the earliest known exclusive marker of pituitary identity, and it is detectable at embryonic day 11.5 (e11.5) in the mouse and rat (*Sornson et al., 1996*; *Yoshida et al., 2009*). Genetic tracing experiments revealed that *Sox2*-expressing cells in the embryonic pituitary give rise to *Sox2*-expressing cells postnatally and to all the hormone-producing cell types of the anterior pituitary gland, demonstrating that SOX2-positive cells are the pituitary stem cells (*Rizzoti et al., 2013*; *Andoniadou et al., 2013*). We recently demonstrated that all the hormone-producing cells of the anterior and intermediate lobes of the pituitary gland pass through a *Prop1* expressing intermediate (*Davis et al., 2016*). Pituitary stem cells are reported to express PROP1 and SOX2 (*Garcia-Lavandeira et al., 2009*), but the overlap in expression of these genes during mouse embryogenesis has not been analyzed. PROP1-expressing cells are largely co-incident with SOX2 expressing progenitors at e12.5, although SOX2-positive cells extend over a larger area of Rathke's pouch (*Figure 1A*, left panel). Later in development, at e14.5, PROP1 expression is decreased, particularly in the dorsal region of Rathke's pouch, where the highly proliferative SOX2-positive cells still predominate. At this time, *Prop1*-expressing cells are predominately located in the transitional zone, where cells are migrating ventrally to populate the anterior lobe (*Figure 1A*, left panel) (*Ward et al., 2005*; *Suh et al., 2002*). One of the known functions of PROP1 is to activate expression of another pituitary-specific transcription factor, POU1F1. Little or no co-localization of POU1F1 and SOX2 is observed in the cell nuclei, either during development or after birth (*Figure 1A*, right panel). The consistently mutually exclusive expression of SOX2 and POU1F1 supports the idea that POU1F1 regulates a later stage of differentiation, and it suggests that any effect of POU1F1 on pituitary stem cells would be indirect. Thus, PROP1, but not POU1F1, is initially expressed in SOX2 expressing stem cells in the embryonic pituitary, and the co-localization of PROP1 and SOX2 is transient, ceasing as cells migrate and transition to differentiation.

### PROP1 is necessary for progenitor proliferation and expression of transitional cell markers

The progenitor cells in Rathke's pouch exit the cell cycle to undergo differentiation. Non-cycling precursors are restricted both spatially and temporally during normal development and are marked by expression of p57$^{Kip2}$ and cyclin E. In most tissues, cyclin E marks the transition from G1 to S phase, but it marks non-cycling precursors in the developing pituitary gland (*Bilodeau et al., 2009*). As differentiation proceeds, p27$^{Kip1}$ is up-regulated concomitant with extinguishing p57$^{Kip2}$ expression, and p27$^{Kip1}$ protects cells from re-entering the cell cycle (*Bilodeau et al., 2009*). To investigate whether PROP1 has a role in regulating the progression of proliferating stem cells to a transition phase, we evaluated the expression of transitional cell markers in *Prop1* loss-of-function mutant (*Prop1*$^{df/df}$, p.Ser83Pro) and control pituitaries. Immunofluorescence revealed CYCLIN E expression in the expected transitional zone of non-cycling progenitors in normal e13.5 pituitaries, but it is completely absent at e13.5 in the pituitaries of *Prop1*$^{df/df}$ mice (*Figure 1B*). Immunoreactivity for

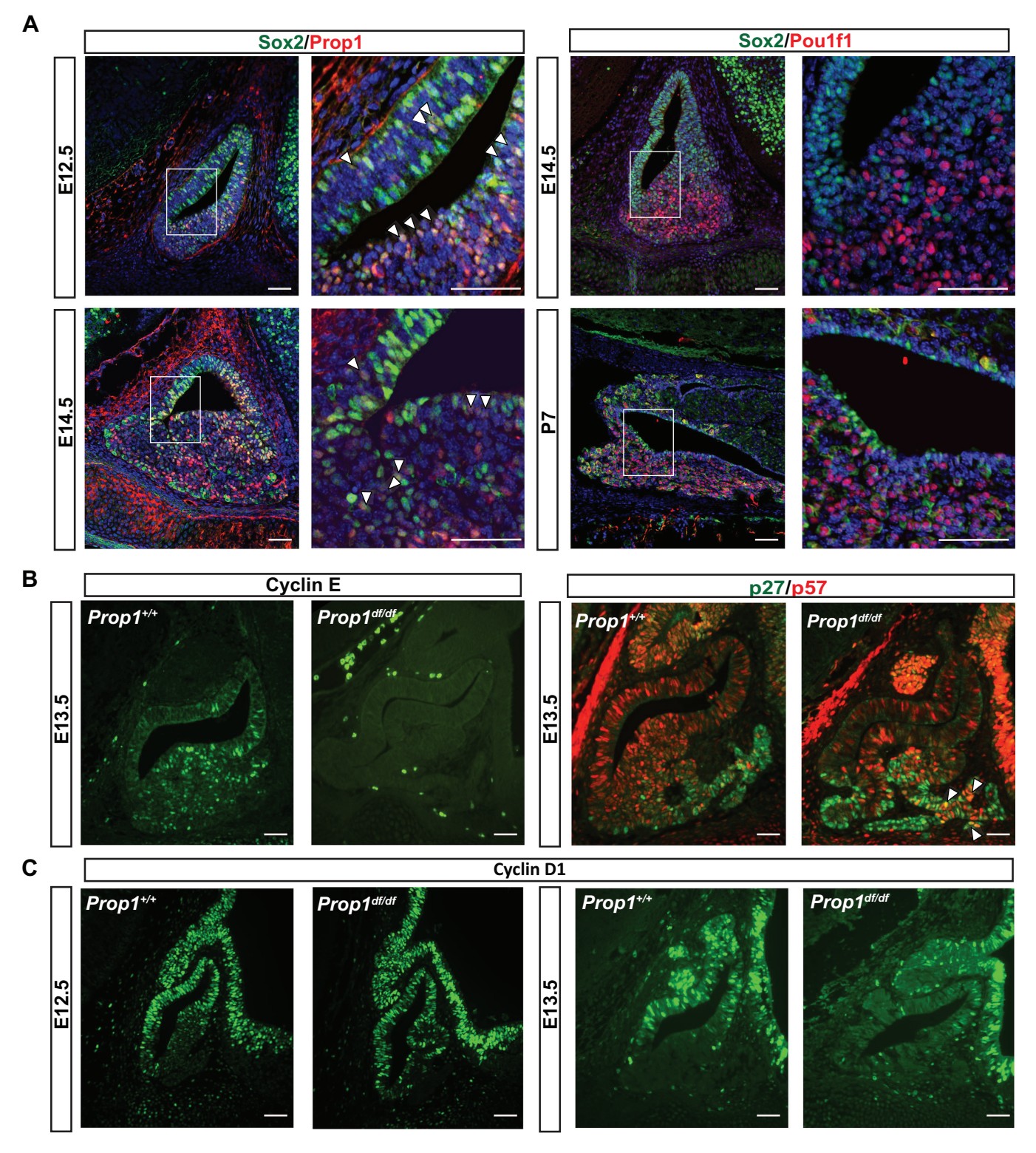

**Figure 1.** PROP1 is expressed in pituitary stem cells and is necessary for cell cycle regulation. (**A**) Left Panel: Double immunofluorescence reveals co-staining of PROP1 (red) and SOX2 (green) at e12.5 in the dorsal and ventral areas of Rathke's pouch (arrowheads). At e14.5 less co-localization was observed (arrowheads). Right Panel: Nuclear expression of SOX2 (green) and POU1F1 (red) is mutually exclusive during development (e14.5) and afterbirth (P7). Cell nuclei were stained with DAPI (blue). (**B**) Immunofluorescence was performed on embryonic sections of *Prop1df/df* and normal controls at e13.5 with a primary antibody for CYCLIN E (green). No CYCLIN E expression was detected in the developing pituitary glands of mutants. At

*Figure 1 continued on next page*

*Figure 1 continued*

e13.5 there are more cells double positive for p27^kip1 (green) and p57^kip2 (red) in the *Prop1*^dfdf relative to *Prop1*^+/+ (arrowheads). (**C**) Immunofluorescence reveals no changes in CYCLIN D1 expression at e12.5 but a decreased level later in development of *Prop1*^df/df pituitaries compared to controls. Scale bars 100 μm.

p57^kip2 and p27^kip1 was spatially and temporally normal in *Prop1*^df/df pituitaries, but the number of double positive cells was abnormally high in mutants (*Figure 1B*). This suggests that *Prop1* deficiency causes an abnormal progression from stem cell to differentiated cell. Cyclin D1 is expressed during the G1 phase of the cell cycle and is essential for cells to passage into S phase. At e12.5, CYCLIN D1 is expressed mainly in the proliferative zone of wild-type and *Prop1* mutant pituitaries. However, at e13.5, there is a reduction in CYCLIN D1-positive cells in dwarf pituitaries (*Figure 1C*). These results show that *Prop1* is necessary for several aspects of cell cycle regulation during embryogenesis: promoting proliferation of progenitor cells marked by Cyclin D1, transitioning them out of the cell cycle to express Cyclin E, and progressing from p57^kip2-positive transitional cells to p27^kip1-positive differentiating cells.

## PROP1 is required to maintain normal SOX2 expression after birth

The rodent pituitary gland undergoes two distinct waves of cell proliferation and differentiation, one occurring during embryogenesis and a second one during the first 3 weeks afterbirth in the mouse (*Gremeaux et al., 2012*; *Zhu et al., 2007*; *Carbajo-Pérez and Watanabe, 1990*). The known pattern of *Prop1* expression correlates with the first wave of proliferation, which peaks at e12.5 and wanes at e14.5, but expression during the postnatal wave of cell proliferation has not been investigated. Using qRT-PCR, we discovered high *Prop1* mRNA levels at postnatal days 3 and 7 (P3 and P7), that are similar to the peak levels at e12.5 and coincident with the second wave of cell proliferation (*Figure 2—figure supplement 1*). We also used qRT-PCR to assess the temporal expression patterns of *Prop1*-dependent genes (*Pou1f1* and *Notch2*), and the stem cell markers *Sox2*, *Gfra2* and *Sox9* during these waves of pituitary growth (*Figure 2—figure supplement 1*). We found that *Prop1* and all these genes are expressed during the postnatal wave of pituitary expansion, and their mRNA levels are at higher or similar levels to those found in embryonic pituitaries.

To investigate whether *Prop1* has novel roles in the postnatal pituitary expansion phase beyond of its regulation of *Pou1f1*, and based on our previous observation where PROP1 and SOX2 are co-expressed on the same cells, we performed qRT-PCR assays for stem cell markers in *Prop1*^df/df and *Pou1f1*^dw/dw (p.Trp251Cys) P7 pituitaries. *Prop1* mutants have higher *Sox2* mRNA levels compared to their wild-type littermates, but *Sox2* mRNA levels were not significantly altered in *Pou1f1* mutant pituitaries (*Figure 2A*). Despite the elevation in *Sox2* expression in *Prop1* mutants, the mRNA levels of *Sox9* and *Gfra2* were not affected, implying heterogeneity within the stem cell population or at least independent regulation of accepted stem cell markers by *Prop1* (*Figure 2A*). Also, immunofluorescence performed on pituitary sections revealed an increase in SOX2-positive cells in the marginal zone of *Prop1* mutants in comparison to both *Pou1f1* mutants and control mice (*Figure 2B*). At P7, pituitaries from *Prop1*^df/df mice have a dysmorphic appearance because progenitors fail to migrate away from the marginal zone, and the overall size of the mutant gland is reduced (*Ward et al., 2005*). Pituitaries from *Pou1f1*^dw/dw mice are also smaller than normal at this age, but they have normal morphology (*Ward et al., 2005*; *Suh et al., 2002*). We assessed whether expression of CYCLIN E is affected by *Prop1* or *Pou1f1* deficiency during the period of postnatal pituitary growth. Using real-time PCR, we observed that mRNA levels of *CyclinE* were decreased in P7 pituitaries from both mutants relative to their wild-type littermates (*Figure 2C*). The increase in SOX2-positive cells around the abnormal marginal zone and the failure to express CYCLIN E in *Prop1* mutants is consistent with a role of *Prop1* in suppressing stem cell fate and stimulating progenitor cell migration during the second wave of pituitary expansion in postnatal pituitaries.

To further analyze the expression of *Prop1* and stem cell markers postnatally, we performed double immunohistochemistry using antibodies against PROP1, SOX2 and SOX9. The missense mutation in *Prop1* mutant mice does not lead to enhanced protein degradation (*Gage et al., 1996*), and the mutated protein can be detect using the PROP1 antibody directed against the N terminus. We characterized the specificity of the PROP1 antibody by performing immunofluorescence for PROP1 on e12.5

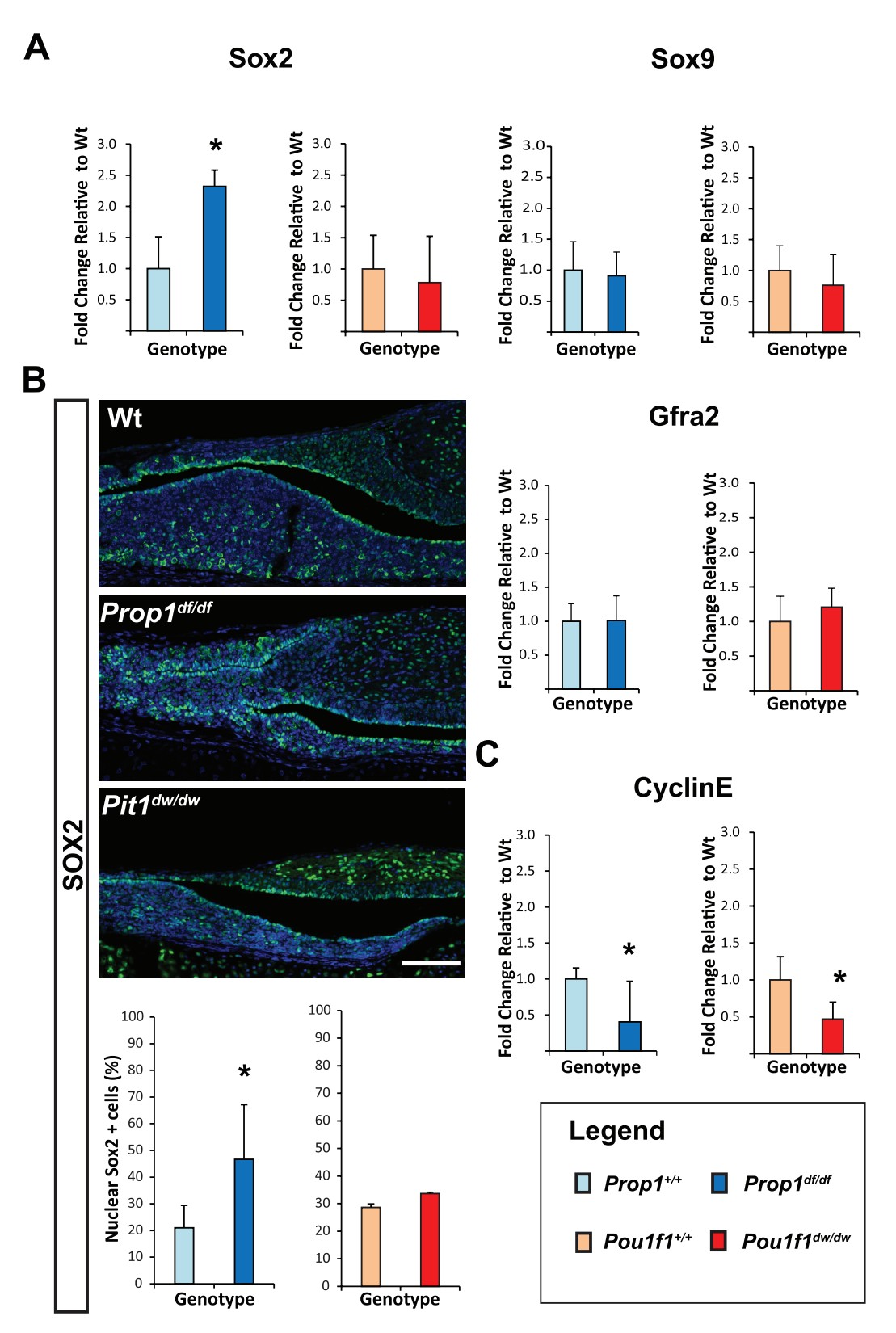

**Figure 2.** Loss of *Prop1* but not *Pou1f1* results in increased SOX2 expression at P7. (**A**) *Sox2* mRNA levels are increased in whole pituitaries of *Prop1*df/df mice at P7 relative to controls. No changes were observed in *Pou1f1* mutant P7 pituitaries. *Sox9* and *Gfra2* mRNA levels are similar in whole pituitaries

*Figure 2 continued on next page*

*Figure 2 continued*

of *Prop1*df/df, *Pou1f1*dw/dw and control pituitaries at p7 (N = 3). Samples were normalized to GAPDH. One-way ANOVA (OWA) and * indicates p<0.05 relative to control. (B) SOX2 is expressed in postnatal pituitaries (P7) in cells lining the pituitary cleft and also in the parenchyma of the anterior lobe in mutant and control mice. Immunofluorescence detection of SOX2 (green) was performed on coronal pituitary sections of wild type, *Prop1*df/df and *Pou1f1*dw/dw mice. An increase in SOX2 expression was detected only in *Prop1* mutant pituitaries. Cell nuclei were stained with DAPI (blue). Scale bars: 100 μm. The charts represent the percentage of cells that immunostain for the indicated marker, N = 5. One-way ANOVA (OWA) and * indicates p<0.05 relative to control. (C) *Cyclin E* mRNA levels are decreased in *Prop1*df/df and *Pou1f1*dw/dw P7 pituitaries (N = 5). Samples were normalized to GAPDH. OWA and * indicates p<0.05 relative to control.

The following figure supplement is available for figure 2:

**Figure supplement 1.** Temporal expression patterns of Prop1, PROP1 targets and stem cell markers during pituitary development.

embryos from wild-type and *Prop1*-/- mice, which carry a deletion in the *Prop1* gene and are not expected to produce a stable protein (*Figure 3—figure supplement 1*) (*Nasonkin et al., 2004*). As expected, strong, nuclear PROP1 immunoreactivity is detected throughout the middle zone of Rathke's pouch in wild-type embryos. No PROP1 staining was detected in the pituitaries of *Prop1*-/- mice, confirming the specificity of the antibody (*Figure 3—figure supplement 1*). The expansion of SOX2-positive cells in *Prop1*df/df pituitaries correlates with an expansion of *Prop1*-expressing cells in the marginal zone (*Figure 3A and C*). The expanded population of SOX2-expressing cells in *Prop1*df/df dwarf pituitaries exhibit less co-localization with SOX9 relative to controls (*Figure 3B and D*). Overall, these results suggest that *Prop1* is crucial to promote stem cell engagement in the postnatal pituitary.

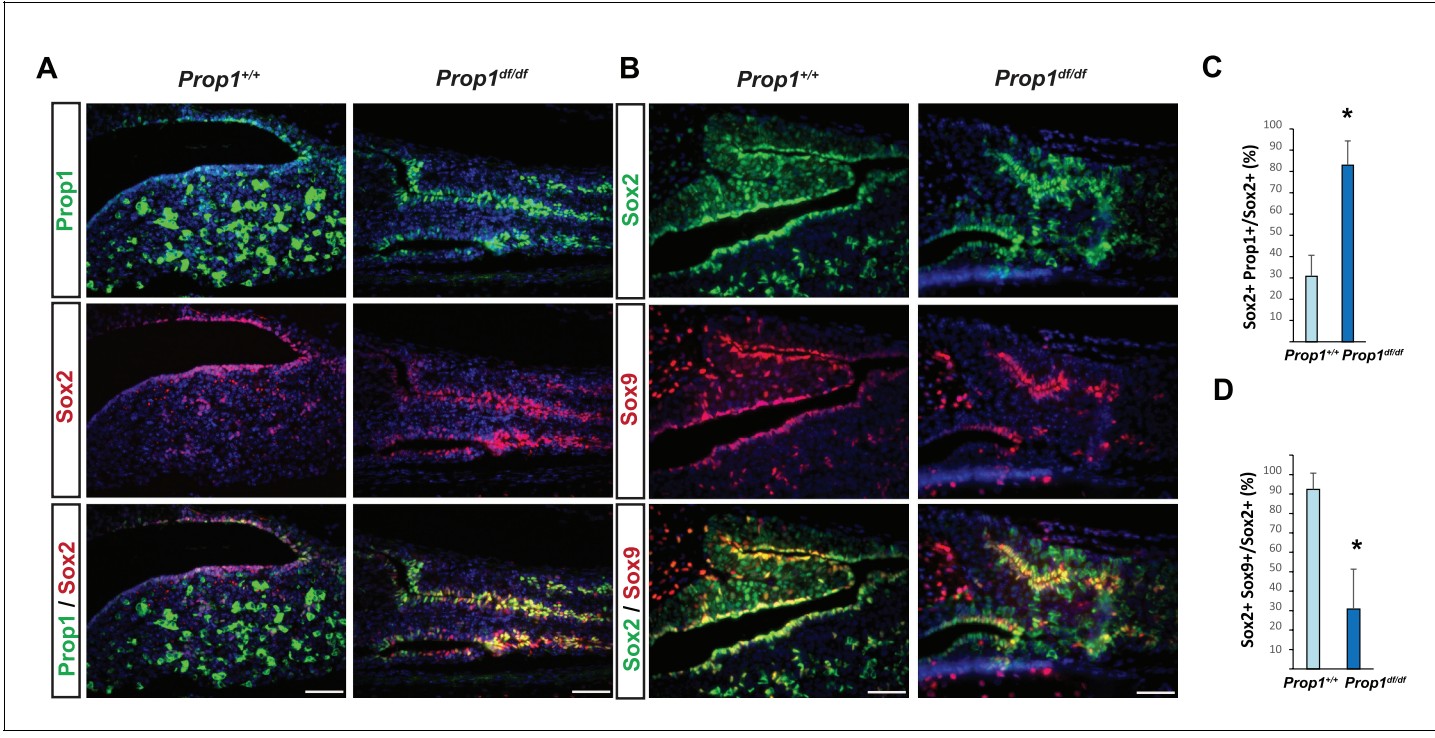

**Figure 3.** *Prop1* regulates Sox2+ stem cell population. Double immunofluorescence labeling of PROP1 and SOX2 (A) and SOX2 and SOX9 (B) in wild-type control and *Prop1* mutants at P3. (C) Quantification SOX2+, PROP1+ cells among SOX2+ cells in the marginal zone in control and mutants. (D) Quantification SOX2+, SOX9+ cells among SOX2+ cells in the marginal zone in control and mutants. Data are represented as mean ± SEM (n = 3 mice, *p<0.05). Scale bars, 50 μm.

The following figure supplement is available for figure 3:

**Figure supplement 1.** Immunofluorescence for PROP1 on e12.5 sagittal sections.

## *Prop1* affects pituitary stem cell properties

Adherent colonies arising from single stem cells can be cultured from postnatal and adult pituitaries in vitro. These colonies represent a mixture of stem cells and cells that have engaged in the transitioning process to differentiation (*Andoniadou et al., 2013*). We analyzed the number of colony-forming cells (CFCs) per pituitary in normal and mutant mice postnatally, as a measure of the stem cell reserve and engagement (*Gaston-Massuet et al., 2011*). We demonstrated that both mutants, *Prop1*$^{df/df}$ and *Pou1f1*$^{dw/dw}$, can generate progenitor CFCs in culture that express stem cell markers, such as SOX2, SOX9, GFRA2 and CYCLIN D2 by immunostaining (*Figure 4—figure supplement 1*). The number of CFC per pituitary at postnatal days 7, 13 and 38 is reduced in *Prop1* mutant relative to wild-type pituitaries (*Figure 4A*, left panel). Colonies from *Prop1*$^{df/df}$ mice differ qualitatively from those of normal mice. Wild-type mice produce colonies that have two visually distinct types. About half of the colonies are comprised of cells that are compact and piled up, which is typical for undifferentiated progenitors (*Gaston-Massuet et al., 2011*). The rest of the wild-type colonies are comprised of large, flat cells, but 100% of *Prop1* mutant colonies are the flat cell type (*Figure 4B and C*, left panel). After 4 days in culture, the difference in morphology in the *Prop1* mutant colonies is already apparent (*Figure 4D*). In contrast to *Prop1*$^{df/df}$, the pituitaries from *Pou1f1*$^{dw/dw}$ mice produced similar number of colonies per pituitary gland at all ages analyzed relative to wild types, and their morphology was representative of wild type (*Figure 4A–C* right panels). Together, these studies demonstrate that *Prop1* is necessary to maintain normal stem cell pools during postnatal pituitary maturation and adulthood.

## *Prop1* regulates expression of EMT pathway genes in stem cell colonies

In order to discover the mechanism whereby *Prop1* deficiency uniquely affects the population of stem cells in the postnatal pituitaries, RNA-Seq experiments were performed on stem cell colonies originating from P13 pituitaries from *Prop1*$^{df/df}$, *Pou1f1*$^{dw/dw}$ and their respective wild-type littermates. Three independent samples of total RNA for each experimental group were collected and analyzed using RNA-Seq technology. A comparison of the gene expression differences in wild types and mutants revealed that the *Prop1* mutation had the greatest effect on gene expression in stem-cell-derived colonies. A total of 2035 genes were differentially expressed (p value $\leq$ 0.05, |log2FC| $\geq$ 1) in colonies from *Prop1* mutants compared with their wild-type littermates, while only about 1/3 that number were differentially expressed in *Pou1f1* mutants compared to their wild-type littermates (752 genes; p value $\leq$ 0.05, |log2FC| $\geq$ 1). The two mutants share a total of 386 genes that are differentially expressed relative to their wild-type littermates (p value $\leq$ 0.05, |log2FC| $\geq$ 1), revealing a set of common effects on colony-forming properties of postnatal pituitary progenitors (*Figure 5A*).

We carried out a pathway analysis to determine which major categories of gene expression were uniquely affected by *Prop1* deficiency in stem-cell-derived colonies. Remarkably, the gene expression profile supports the idea that *Prop1*-mutant-derived colonies fail to undergo an Epithelial-Mesenchymal Transition (EMT)-like process that occurs in both wild-type and *Pou1f1*$^{dw/dw}$ colonies. The epithelial markers E-cadherin (*Cdh1*) and claudin-23 were uniquely up-regulated in *Prop1* mutant colonies. Moreover, markers of EMT including the regulatory gene *Zeb2*, which is a transcriptional repressor of the *Cdh1* gene, are expressed at lower levels only in *Prop1*$^{df/df}$ colonies (*Figure 5A*). We detected up-regulation of miR-200a in colonies from *Prop1* mutant pituitaries. This microRNA inhibits the accumulation of ZEB proteins at the *Cdh1* gene promoter (*Vandewalle et al., 2005*). Other genes in the EMT-like pathway that exhibited altered expression include the matrix metallopeptidases (*Mmps*), a disintegrin and metalloproteinases with or without thrombospondin motifs (*Adam* and *Adamts*). Moreover, some of the signaling pathways that are down-regulated in colonies from *Prop1* mutants are well known for being EMT-inducers, including those mediated by Notch, transforming growth factor β (TGFβ), bone morphogenetic protein (BMP), Wnt and SHH signaling (*Figure 5A*).

We validated differential expression of selected genes in stem cell colonies from *Prop1* mutant and wild-type pituitaries. A qRT-PCR analysis showed that *Cdh1* and *Claudin 23* were elevated, and *Mmp2, Mmp3, Mmp16, Zeb2* and *Gli2* were decreased in the *Prop1* mutant colonies relative to colonies from wild-type littermates (*Figure 5B*). Expression of CDH1 was also analyzed by immunohistochemistry. Colonies from *Prop1*$^{df/df}$ pituitaries showed increased in CDH1-positive cells (*Figure 5B*). *CyclinE* was decreased in *Prop1* mutant colonies, at mRNA and protein levels,

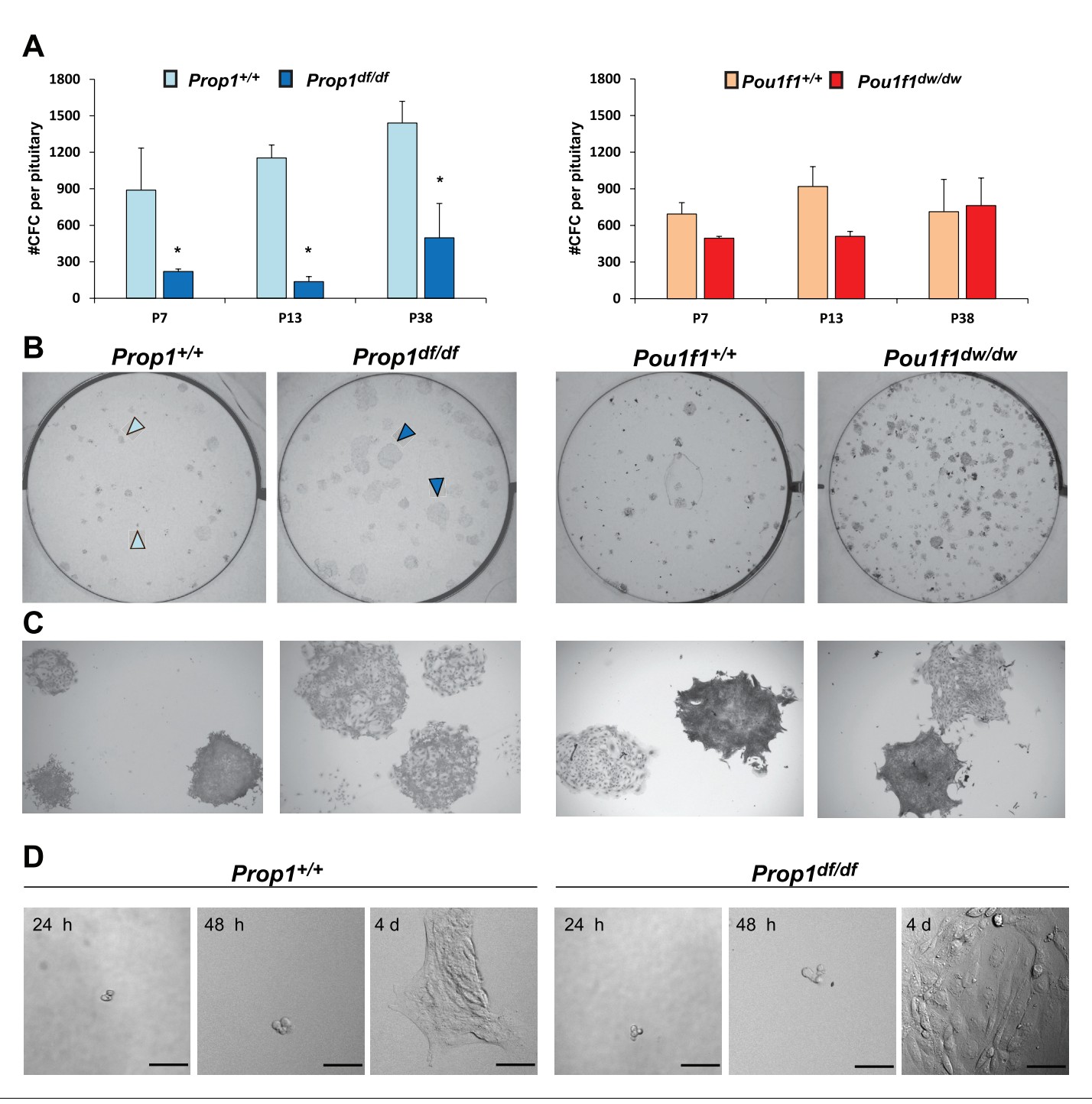

**Figure 4.** *Prop1* regulates pituitary stem cells after birth. (**A**) Number of colony-forming cells per pituitary after 10 days in culture (N = 3–6 pituitaries per stage). Two-way ANOVA *p<0.05 for *Prop1*<sup>df/df</sup> vs aged matched *Prop1*<sup>+/+</sup> littermates, *post hoc* Fisher analysis. Plots denote the mean ± SEM and * indicates p<0.05 relative to control. (**B** and **C**) Pictures of the tissue culture plates showing colonies from *Prop1* (left) and *Pou1f1* (right) mutant pituitaries fixed and stained, show the different morphology of colonies from *Prop1*<sup>df/df</sup> pituitaries (arrowheads). (**C**) Photos taken at higher magnification, 2x. (**D**) Time lapse microscopy shows the formation of a colony from a single stem cell captured 24 hr after the cells were plated and 4 days later. Colonies from *Prop1* mutant pituitaries have an altered morphology since the colony formation starts around day 4. Scale bar 100 μm. See also *Figure 4—figure supplement 1*.

The following figure supplement is available for figure 4:

*Figure 4 continued on next page*

*Figure 4 continued*

**Figure supplement 1.** Pituitary stem-cell-derived colonies from *Prop1* and *Pou1f1* mutant pituitaries.

consistent with our findings in pituitary tissue (*Figure 5B*). Because *Zeb2* is an inducer of EMT in other tissues (*Vandewalle et al., 2005*), we tested whether reduction in *Zeb2* expression was sufficient to alter colony morphology and gene expression. We achieved effective reduction of *Zeb2* expression with siRNA designed to target *Zeb2,* but not with the siRNA negative control (*Figure 6*). Lowered *Zeb2* expression was sufficient to reduce expression of *Gli2* and increase expression of *Cdh1.* Treatment with *Zeb2* siRNA appeared to block formation of tightly packed colonies typical of normal pituitary stem cell cultures (*Figure 6A and B*). Thus, in the absence of *Prop1* or *Zeb2* the colonies fail to initiate the EMT-like process and exhibit abnormal morphology and EMT marker gene expression. This supports the idea that *Prop1* induces EMT-like process in the stem cell population.

We examined expression of the EMT inducer *Zeb2* and cadherins during pituitary development in control and *Prop1*$^{df/df}$ mice (*Figure 7A*). At e14.5, *Zeb2* is normally expressed in cells in the ventral part of the Rathke's Pouch (*Figure 7A*, between the two lines), and this zone of enriched expression is almost absent in *Prop1* mutant pituitaries. CDH1 is normally expressed throughout the Rathke's pouch, but CDH1 staining is stronger in the ventral part of the *Prop1* mutant pouch, particularly along the marginal zone (*Figure 7A*, box and higher magnification image), although there was no obvious difference in expression of the mesenchymal marker, N-CADHERIN or CDH2. Next, we explored CDH1 expression at P13, the time point at which the CFCs were collected for subsequent RNA-Seq analysis. We found that *Prop1*$^{df/df}$ mutants have abnormal expression of CDH1 in cells distant from the marginal zone. Taken together, these results show that *Prop1* is necessary for suppressing SOX2 expression in stem cells and for driving them to undergo an EMT-like process critical for populating the anterior lobe with progenitors capable of differentiating into the PIT1 lineage.

## Identification of putative *Prop1* target genes by ChIP sequencing

*Pou1f1* target genes have been explored (*Skowronska-Krawczyk et al., 2014*), but only a few direct *Prop1* target genes are known (*Olson et al., 2006*). PROP1 binds DNA near the *Hesx1* and *Pou1f1* genes and suppresses or activates their expression, respectively (*Sornson et al., 1996*). Little is known about other direct targets of PROP1 regulation. There are no available cell lines that express *Prop1,* and chromatin immunoprecipitation (ChIP) analysis is difficult in small, developing tissues. To identify potential *Prop1* target genes, we engineered expression of a biotin-tagged PROP1 in pituitary GHFT1 cells, a cell line that was originally developed by targeted oncogenesis in transgenic mice using the *Pou1f1* enhancer and promoter sequences to drive SV40 T-antigen expression (*Lew et al., 1993*). GHFT1 cells are thought to represent immortalization of cells in the pituitary primordium at approximately e14.5 when *Prop1* has activated *Pou1f1,* but the hormone genes that are downstream targets of *Pou1f1* are still silent. We generated a stable, derivative, GHFT1 cell line that expresses the bacterial biotin ligase BirA (BirA) using neomycin selection, and a second, stable derivative using puromycin drug selection that expresses BirA and recombinant PROP1 (Prop1Tag). *Prop1* was engineered to encode a 23 amino acid BirA recognition motif fused to the amino terminus (*Figure 8A*) (*Yu et al., 2009*). We chose to modify the amino terminus of PROP1 because is not well conserved (*Ward et al., 2007*). Western blot analysis, using a BirA antibody, demonstrated expression of BirA in both the BirA cell line and BirA-Prop1Tag cell line. Using streptavidin horseradish peroxidase, we demonstrated in vivo biotinylation of the recombinant PROP1 in the BirA-Prop1-Tag cell line (*Figure 8B*).

To confirm that the Prop1Tag protein has normal function, we generated transgenic mice expressing the Prop1Tag construct, and carried out the appropriate crosses with *Rosa26*$^{BirA/BirA}$ transgenics and *Prop1*$^{+/-}$ mice to generate *Prop1*$^{Tag}$, *Rosa26*$^{BirA}$, *Prop1*$^{-/-}$ mice. Mice with the latter genotype grew normally, confirming that biotinylated PROP1 is active and functional and able to correct the dwarfism characteristic of *Prop1*$^{-/-}$ mutants (*Figure 8—figure supplement 1*). The expression of *Pit1* and pituitary dysmorphology were also rescued during development by the *Prop1Tag* transgene (*Figure 8—figure supplement 1*).

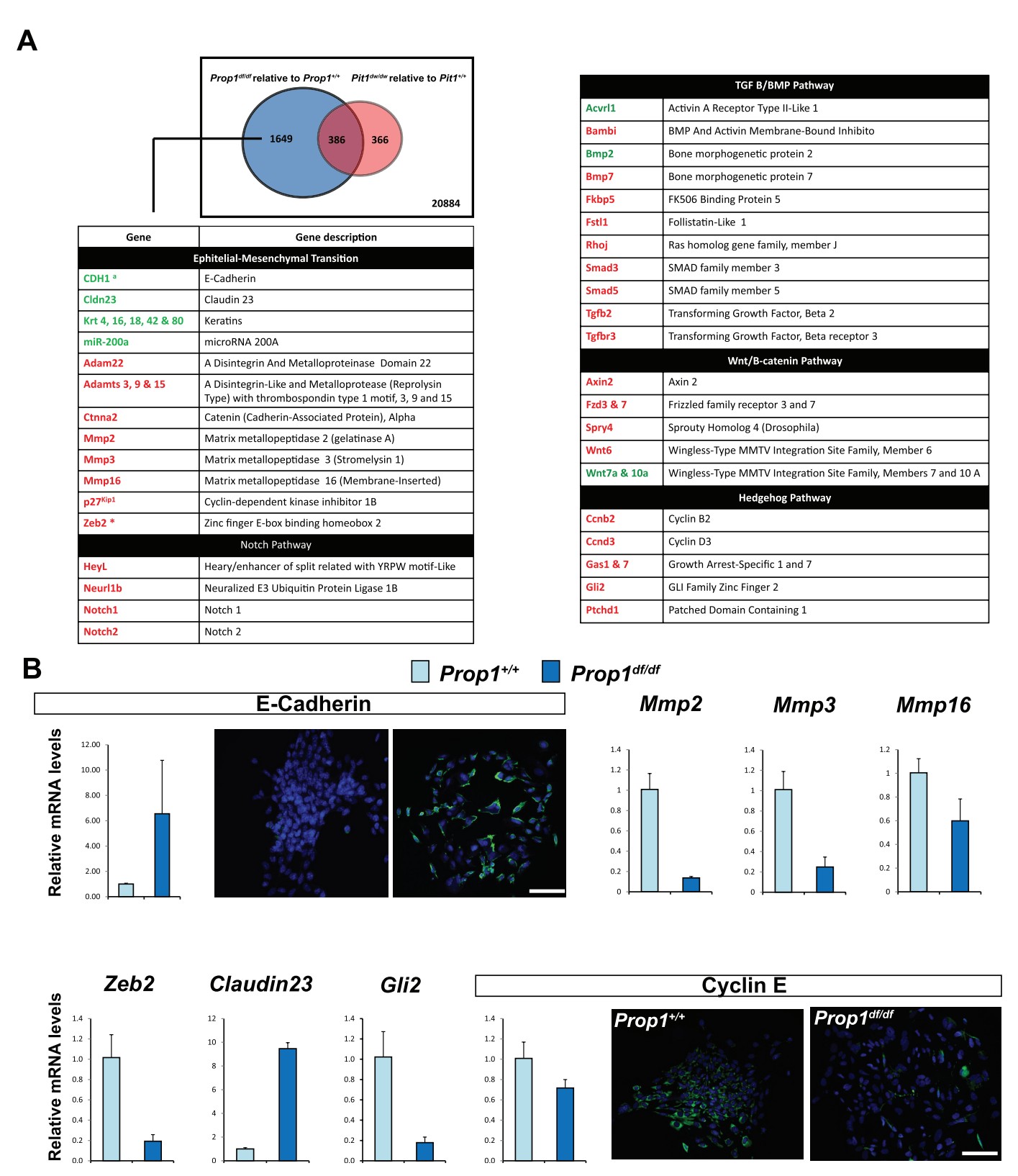

**Figure 5.** Differential gene expression in *Prop1*[df/df] and *Pou1f1*[dw/dw] pituitary stem-cell-derived colonies at P13. (**A**) A Venn diagram illustrates the total number of genes with significantly different expression in *Prop1* mutants relative to their wild-type littermates (blue circle) and *Pou1f1* mutants relative

*Figure 5 continued on next page*

*Figure 5 continued*

to their wild-type counterparts (red circle). The table shows examples of genes up regulated (green) or down regulated (red) (p value $\leq$ 0.05, |log2FCl $\geq$ 1) uniquely in the *Prop1* mutant colonies relative to *Pou1f1* mutants. (B) RT-qPCR validation of RNA-Seq data. GAPDH was used as an internal control (N = 4) OWA and * indicates p<0.05 relative to control. Immunostaining for CDH1 shows an increased in protein expression on colonies from *Prop1$^{df/df}$* pituitaries compare to control. CYCLIN E immunostaining reveals that *Prop1$^{+/+}$* colonies express this protein but is absent on *Prop1$^{df/df}$* colonies. Cell nuclei were stained with DAPI (blue). Scale bar 100 μm. Plots denote the mean ± SEM. qPCRs were done using at least three technical replicates.

Streptavidin (SA)-based ChIP was performed on GHFT1 cells containing BirA alone or in combination with Prop1Tag. This takes advantage of the extremely high affinity of avidin for biotinylated proteins (*Viens et al., 2004*). A quantitative DNA-PCR assay was used to assess enrichment for occupancy of Prop1Tag at a known PROP1 binding site in the *Pou1f1* gene. To assess nonspecific background PROP1-DNA binding, a quantitative DNA-PCR assay was carried out for a site in the *Hoxd10* promoter, which is not expressed in the pituitary or other structures anterior to the hindbrain (*Figure 8C*). The known PROP1 binding site in *Pou1f1* was highly enriched relative to input and to the controls. The DNA recovered by SA-ChIP was sequenced with a HiSeq 2000 sequencing analyzer and the results analyzed with a standard bioinformatics pipeline. The enrichment profile near the *Pou1f1* gene identified the known PROP1-binding sites at the early enhancer and the promoter, as well as another, more prominent site at -6 kb, demonstrating the efficacy of our approach for identification of novel PROP1-binding sites (*Figure 8D and E*). We carried out sequence motif searches on the ChIP-Seq peaks (center ± 200 bp) that were recovered throughout the genome using HOMER algorithms. This analysis identified a consensus sequence: TAATNNNATTA in 60% of peaks (p = 1e-1599) (*Figure 9A*). This consensus sequence is the first to be experimentally determined based on genome-wide data and is consistent with the paired homeodomain core consensus expected for PROP1 (*Sornson et al., 1996*). Interestingly, additional enriched transcription factor binding motifs were also identified in approximately 30–38% of peaks (*Figure 9A*). The best match for the motifs TGAGTCAT and CGATTAGC are Jun-AP1 and CRX, respectively. Because CRX (cone rod homeobox) is not expressed in the pituitary gland, the CGATTAGC motif is likely bound by a different OTX-like paired homeodomain protein that cooperates with PROP1 to direct transcriptional programs associated with target genes (*Reed et al., 2008*).

To correlate putative PROP1 target genes with altered gene expression observed in vivo and in CFCs, we carried out a pathway analysis on the genes identified by ChIP-seq (*Figure 9B*). Most of the genes were involved in regulation of EMT pathway, consistent with what we observed for gene expression in CFCs. Genes in the Axonal Guidance Signaling and Cell-Cell Junction Signaling pathways were also detected (*Figure 9—source data 1*). We validated the PROP1 Chip-Seq data for selected genes in these pathways: *Gli2*, *Notch2*, *Zeb2* and *Claudin 23*, using GHFT1 BirA-Prop1Tag cells and their controls, and confirmed that the enrichment was PROP1-specific (*Figure 9C–E*). In summary, these data suggest that PROP1 regulates genes involved in the EMT-like process directly. This expands our understanding of the full spectrum of mechanisms by which *Prop1* regulates pituitary stem cells.

## Discussion

Mutations in several transcription factors, including HESX1, PROP1 and POU1F1 cause pituitary hormone deficiency in mice and humans. HESX1 and POU1F1 are direct targets of PROP1 repression and activation, respectively (*Olson et al., 2006*; *Sornson et al., 1996*). POU1F1 stimulates the differentiation of progenitors into thyrotropes, somatotropes and lactotropes (*Bodner et al., 1988*; *Gordon et al., 1993*; *Ingraham et al., 1988*; *Lin et al., 1994*; *Steinfelder et al., 1991*) by organizing target gene enhancer regions in the nuclear matrix, using a mechanism involving MATRIN-3, β-CATENIN, and the AT-rich binding protein, SATB1 (*Skowronska-Krawczyk et al., 2014*). PROP1 deficiency causes similar hormone deficiencies to those caused by POU1F1 mutations, but in addition, it is uniquely associated with progressive hormone loss, highly dysmorphic Rathke's pouch, and poor vascularization (*Ward et al., 2006*). However, the molecular bases for these aspects of PROP1-specific disease pathologies are unknown.

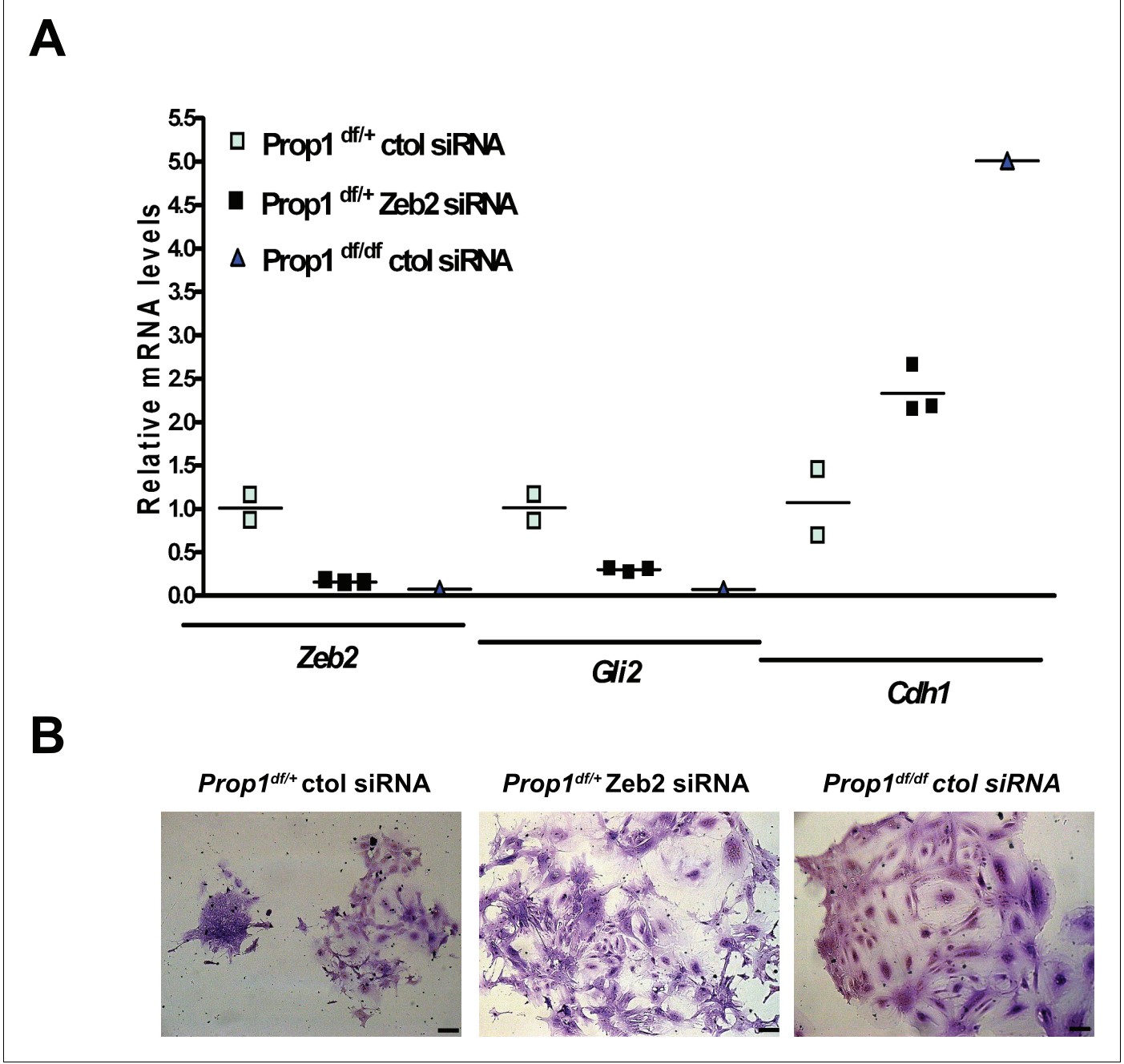

**Figure 6.** Zeb2 knockdown resemble colonies derived from *Prop1* mutant mice. (**A**) *Zeb2, Gli2* and *Cdh1* mRNA levels in colonies from *Prop1*[df/+] and *Prop1*[df/df] pituitaries treated with *Zeb2* siRNA and negative control siRNA (control). Samples were normalized to GAPDH. qPCRs were done using at least three technical replicates. (**B**) Pictures of the colonies from *Prop1*[df/+] and *Prop1*[df/df] pituitaries fixed and stained, show the different morphology of colonies between control siRNA and treated with *Zeb2* siRNA.

Here, we define two new roles of *Prop1*: first, PROP1 is a regulator of pituitary stem cell (SOX2+ cell) population and second, PROP1 triggers an EMT-like process in pituitary stem cells that is crucial for normal pituitary morphology and cell specification of the PIT1 lineage. Our genome-wide analysis of PROP1 binding identifies target genes that regulate these processes. We propose that *Prop1* mediates progenitor cell proliferation by driving cell cycle progression. In the absence of functional *Prop1*, embryonic pituitary progenitors fail to maintain expression of the proliferative markers Cyclin

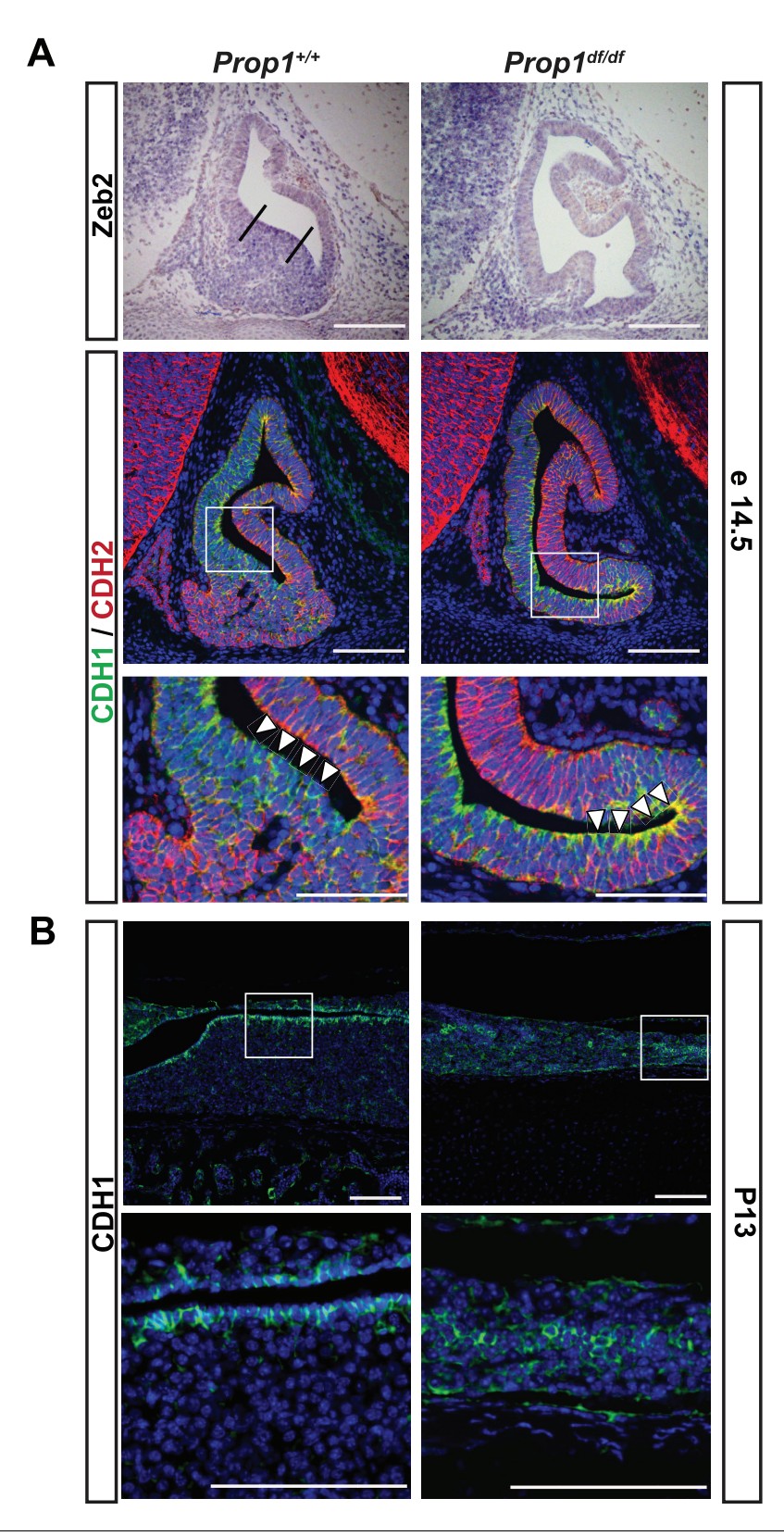

**Figure 7.** Loss of *Prop1* affects EMT pathway genes in pituitary tissues. (**A**) Upper panel:In situ hybridization using a *Zeb2* antisense riboprobe at e14.5 detected expression in the control pituitary. Most of the positive cells are in the ventral area, around the cleft (between the lines). In *Prop1*df/df

*Figure 7 continued on next page*

*Figure 7 continued*

pituitaries very few cells are expressing *Zeb2*. Lower panel: Sagittal sections of 14.5 embryos were stained for CDH1 (green) and CDH2 (red) by immunohistochemistry. In the wild-type pituitary, CDH1 expression is dispersed throughout the Rathke's Pouch, while CDH2 expression is more concentrated in the rostral area and in the forming anterior lobe. In *Prop1* mutant pituitaries, CDH1 expression is increased and concentrated in the area ventral to the lumen, where *Zeb2* expression was absent. White boxes indicate where higher magnification photos were taken. (B) Immunohistochemestry for CDH1 at P13 shows that *Prop1*[df/df] pituitaries have an expanded expression of this endothelial protein in the anterior lobe compared to control pituitaries. White boxes indicate where higher magnification photos were taken. Cell nuclei were stained with DAPI (blue). Scale bar 100 μm.

D1 and Cyclin D2 (*Ward et al., 2005*; *Raetzman et al., 2004*), and they fail to initiate expression of the transitional marker Cyclin E. We did not detect *Prop1*-binding sites near the *CyclinD1, D2* or *E* genes, suggesting that *Prop1* regulation of these genes is indirect. There is a precedent for *Gli2* activating expression of these genes in the developing lung (*Rutter et al., 2010*). *Gli2* is necessary for pituitary progenitor proliferation and normal pituitary morphology (*Wang et al., 2010*). *Gli2*-deficient mice either lack their anterior pituitary or have pituitary hypoplasia (*Wang et al., 2010*). Loss-of-function mutations in the human *GLI2* gene cause pituitary anomalies and holoprosencephaly (*Roessler et al., 2003*). *Gli2* expression is reduced in colonies derived from *Prop1* mutant pituitaries, and PROP1 binds to the 5' region of the *Gli2* gene in cell culture. However, *Gli2* expression is not strictly or directly *Prop1* dependent because *Gli2* is expressed in the mesenchyme surrounding the pituitary early on (at e12.5) and is expressed in pituitary tissue after *Prop1* expression wanes (e15.5).

*Prop1* is required to maintain stem cell pools and stimulate progenitor cells to differentiate. This idea is supported by multiple lines of evidence. First, all the hormone producing cell types of both the anterior and intermediate lobes go through a *Prop1* expressing intermediate (*Davis et al., 2016*). Second, *Prop1* mutants have reduced cyclin D1 and D2 expression in the progenitor compartment embryonically. Third, *Prop1* mutants have more SOX2-positive cells residing in the marginal zone or residual Rathke's cleft, and fewer SOX2-positive cells in the parenchyma, in comparison to both *Pou1f1* mutants and control mice. Fourth, the SOX2-expressing cells in *Prop1* mutants lack SOX9, have reduced colony forming capabilities, and exhibit abnormal colony morphology. PROP1 regulation of *Sox2* expression may be indirect, even if they are expressed in the same cell at one point during development and after birth, since PROP1 did not bind the SOX2 promoter in cell culture.

Another role of PROP1 is to drive an EMT-like process in pituitary stem cells by stimulating expression of *Zeb2*. We determined this using several different approaches: colony-forming assays, in vivo analysis of *Prop1* mutant tissues and genetically engineered cell lines. *Prop1* deficiency affects pituitary colony-forming properties, and the *Prop1* mutant-specific alterations in gene expression in these CFCs indicate failure of an EMT-like process. The classic pattern of EMT involves loss of Cdh1 (E-Cadherin) and up-regulation of Cdh2 (N-Cadherin). In *Prop1* mutants CFCs as well as in pituitary tissues, we observed an increase in Cdh1, however no changes in Cdh2 expression were found. Thus, the process is likely a partial EMT, in which mesenchymal cells do not display all the characteristics of a typical EMT (*Leroy and Mostov, 2007*). A number of transcription factors, including ZEB family members, regulate EMT. PROP1 binds to the *Zeb2* promoter, suggesting the regulation may be direct. The level of *Zeb2* expression is decreased in both CFCs and developmental pituitaries from *Prop1* mutant mice, and reduction in *Zeb2* expression is sufficient to mimic some of the properties of *Prop1*-deficient CFCs. ZEB family proteins reduce epithelial marker expression and increase mesenchymal marker expression during the induction of EMT in other organs (*Peinado et al., 2007*). In addition, *Zeb2* is repressed through microRNA 200 family members (*Brabletz and Brabletz, 2010*), and we found an increase in miR-200a transcripts in colonies from *Prop1* mutant pituitaries. In humans, mutations on the *ZEB2* gene cause Mowat Wilson Syndrome, a disorder associated with mental retardation, microcephaly and short stature (*Mowat et al., 1998*). Taken together, it is tempting to speculate that affected individuals with this pathology could have also pituitary defects.

During development, expansion of organs relies on cell migration and differentiation. These events imply changes in the actin cytoskeleton and altered expression of adhesion molecules and matrix extracellular components. We show altered cell shape, elevated expression of adhesion molecules and reduced levels of matrix metalloproteinases (MMPs) in stem cell colonies from *Prop1*

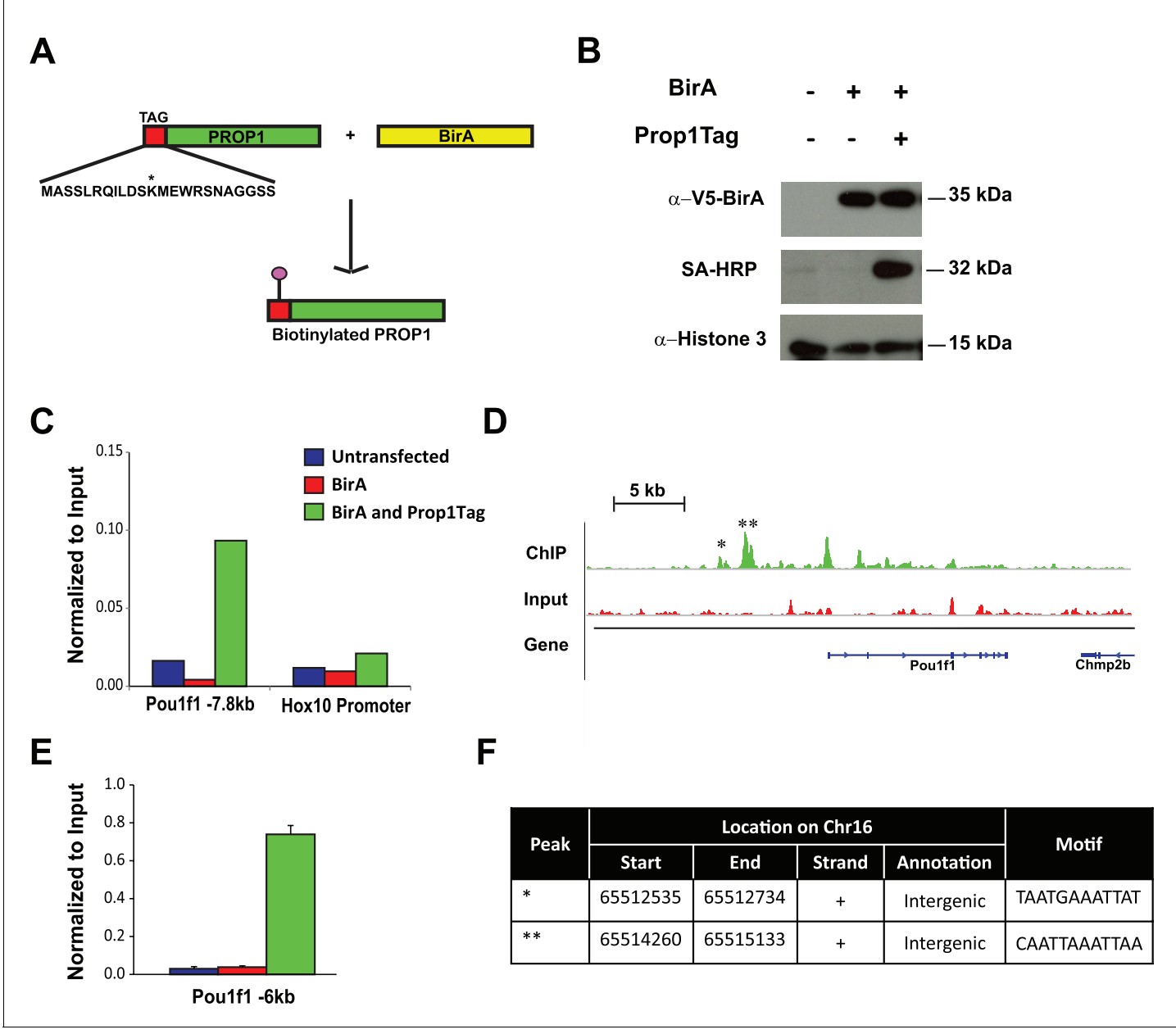

**Figure 8.** Streptavidin ChIP-Seq of biotinylated PROP1 in GHFT1 cells. (**A**) Schematic diagram of biotin tagging system. The *birA* recognition motif tag (red) is shown fused to the amino terminus of PROP1. The biotin acceptor lysine (K) is indicated with *. (**B**) Western blot of protein extracts from GHFT1 cell clones expressing either BirA alone, or BirA and the recombinant Prop1Tag. Upper panel, probed with anti-alpha-V5 antibody for BirA detection; Middle panel, probed with streptavidin horse radish peroxidase (SA-HRP) and lower panel, probed with anti-Histone 3 antibody for protein loading control. (**C**) Quantitative ChIP assay at a known PROP1 occupancy site in the *Pou1f1* promoter and a negative control site (*Hoxd10* promoter) using streptavidin-based ChIP. (**D**) Streptavidin ChIP-Seq enrichment profiles for PROP1 at the *Pou1f1* gene. The known peak at −7.8 kb is indicated with *. A novel peak at -6 kb is indicated with ** and (**E**) validated N = 3 OWA and * indicates p<0.05 relative to controls. Data are mean ± SEM. Each ChIP experiment was repeated at least twice with similar results. (**F**) Table containing coordinates of the peaks at the *Pou1f1* promoter and the motifs found in the peaks.

The following source data and figure supplement are available for figure 8:

**Source data 1.** RNA-Seq performed on GHFT1 BirA and GHFT1 BirA Prop1Tag cells (N = 3).
**Figure supplement 1.** Characterization of *Prop1-biotag* transgenic mice.

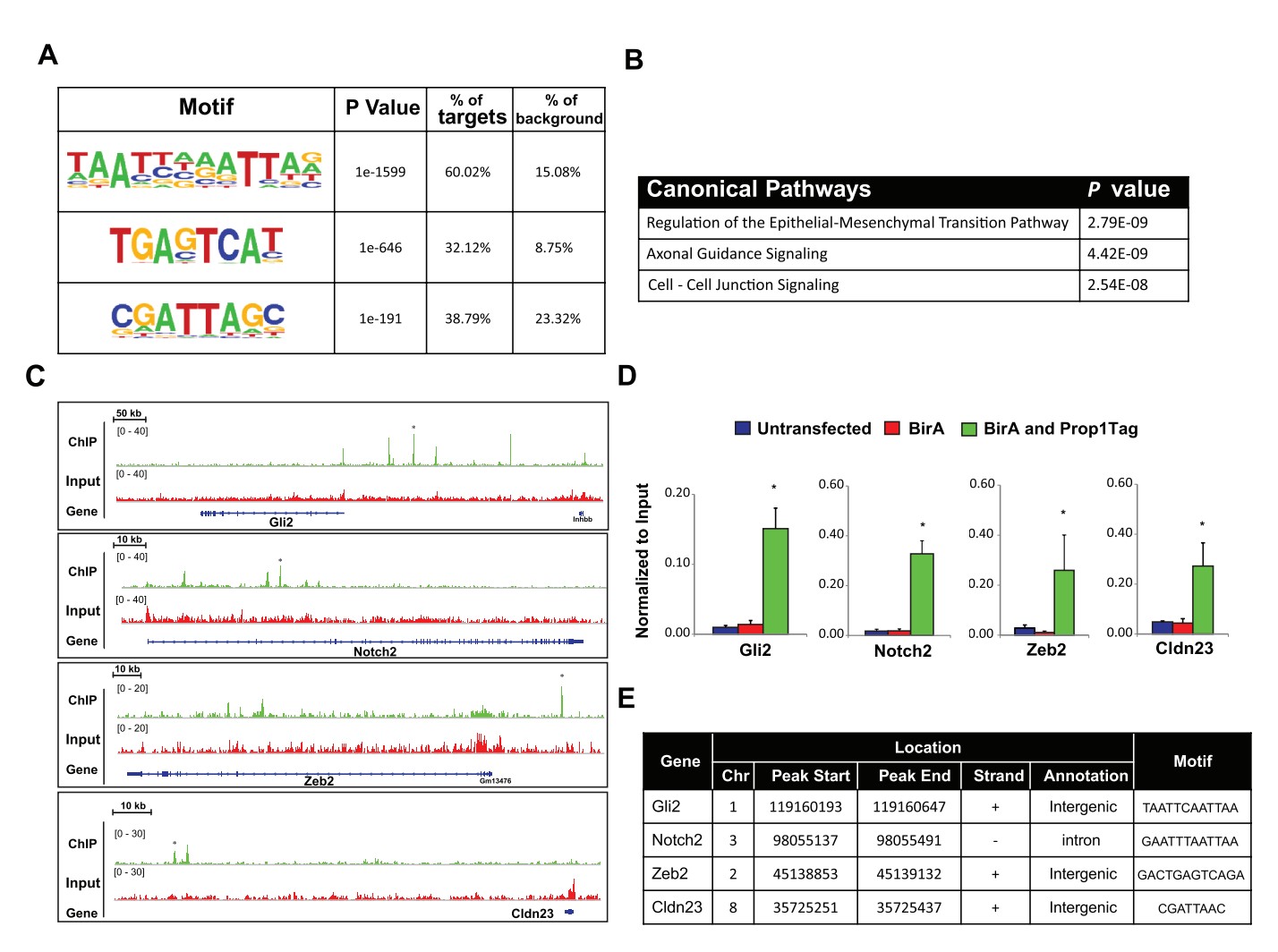

**Figure 9.** PROP1 is a regulator of genes involved in EMT. (**A**) HOMER analysis of new motif enrichment within PROP1 peaks. (**B**) Canonical Pathway Analysis of the putative PROP1 target genes found on the ChIP-Seq. (**C**) Streptavidin ChIP-Seq enrichment profiles. Enrichment peaks corresponding to *Gli2, Notch2, Zeb2* and *Cldn23* are indicated with asterisks. (**D**) Quantitative ChIP assay on candidate promoters. For each gene, primers were designed to amplify region where the PROP1 peak was detected. Note that enrichment of DNA fragments was specific to BirA, Prop1Tag cells, validating the specificity of ChIP-Seq. Experiments were done using three samples for each genotype (N = 3). Data are mean $\pm$ SEM. OWA and * indicates p<0.05 relative to controls. Each ChIP experiment was repeated at least twice with similar results. (**E**) Table containing coordinates of the peaks found on each gene and the enriched motif found by HOMER. See also *Figure 9—source data 1*.

The following source data is available for figure 9:

**Source data 1.** Top list of putative PROP1 target genes that were associated with strong peaks in ChIP-Seq.

mutant mice. MMPs play key roles in embryonic development and are involved in EMT in the morphogenesis of many tissues, including neural crest delamination, endocardial invasion and mammary gland branching formation (*Radisky and Radisky, 2010*). Several genes down regulated in the stem cell colonies from *Prop1* mutant pituitaries were associated with main inducers of the EMT including members of the Notch, TGF-β, Wnt and Hedgehog pathways. All these signaling pathways are also well known for their critical role in pituitary embryonic development and have important roles in tumorigenesis in both humans and mice. EMT plays an important role in cancer pathogenesis, particularly during growth, maintenance and progression of the tumor with local invasion or metastasis (*De Craene and Berx, 2013*; *Pirozzi et al., 2011*; *Mani et al., 2008*). Pituitary adenomas are tumors

with significant prevalence and, although they are often benign, can become invasive (*Melmed, 2011*). At present, the mechanisms underlying the pathogenic processes of initiation, expansion and invasion of pituitary adenomas are still unknown. Our data provide insight into a role of EMT in pituitary development that could be applied in the context of pituitary tumor formation and to other tissues that contain stem cells that undergo an EMT-like process before differentiating, such as during heart and mammary development (*Kim et al., 2014*).

In summary, we showed that *Prop1* is required for normal stem cell pools during embryogenesis and postnatal pituitary expansion. Our genome-wide analyses support the idea that *Prop1* promotes the transition of progenitors to differentiation by inducing expression of EMT drivers like *Zeb2*. In addition, *Prop1* activates expression of cyclin E, a marker of the transition state, and *Pou1f1*, a

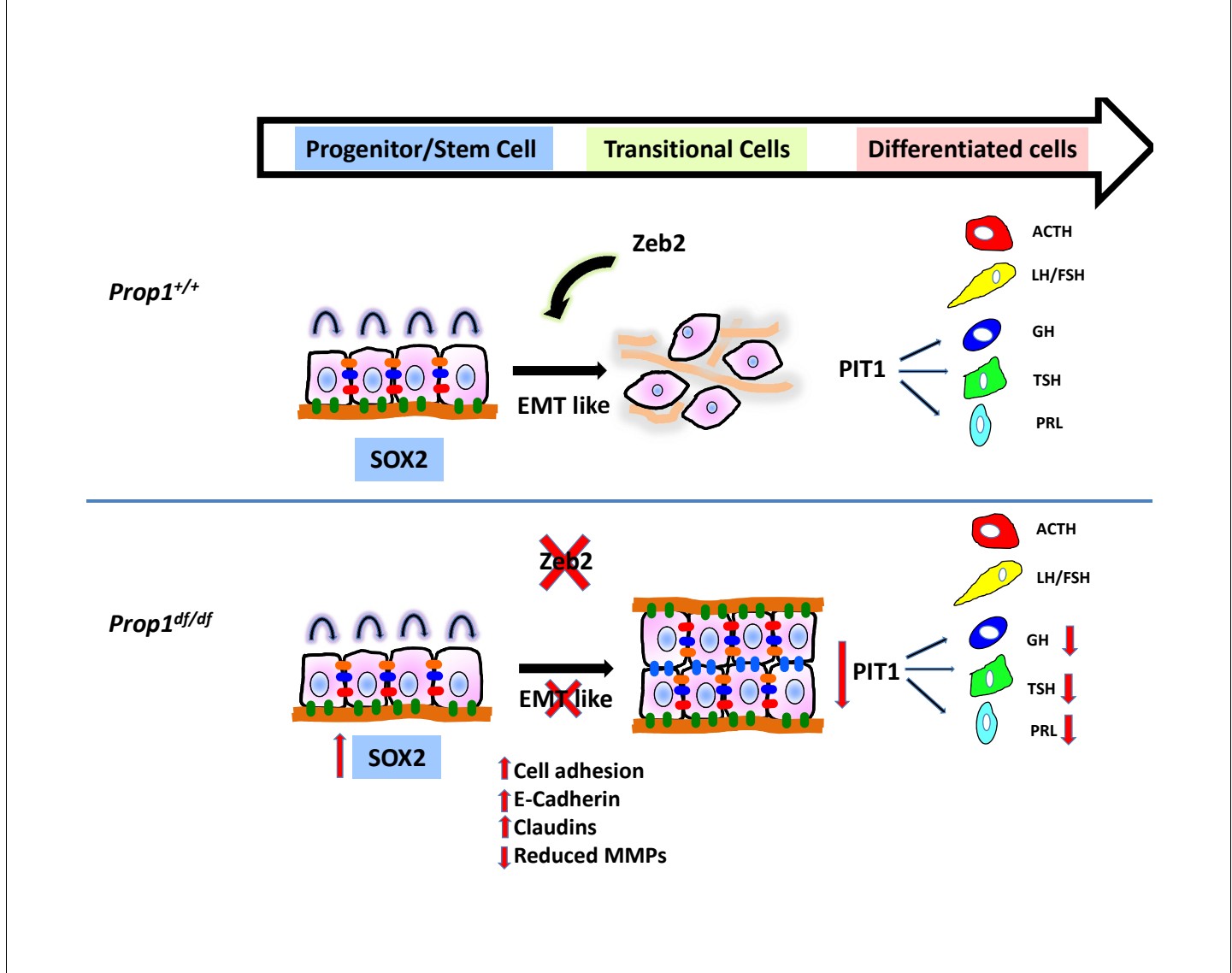

**Figure 10.** Model of *Prop1's* role in the transition of stem cells to differentiation. During normal pituitary development when stem cells transition toward differentiation they exit the cell cycle and express Cyclin E. Our results suggest that for progenitors to differentiate they need to go through an EMT-like process where E-cadherin is down-regulated and the cells lose adhesion. In the absence of *Prop1*, the expression of genes that can induce EMT, like *Zeb2*, is reduced, leading to increased cell adhesion and increased expression of tight junction proteins like claudins. Our data suggest that PROP1 is required for progenitors to progress to the transitional stage marked by Cyclin E expression embryonically, and in the absence of *Prop1*, *Sox2* expression is elevated. The failure of progenitors cells to advance to the transitional stage leads to pituitary hormone deficiency and organ dysmorphology.

marker of the differentiation state (*Figure 10*). This study establishes the mechanism of PROP1 action in pituitary progenitor cells, offers new candidate genes for cases of pituitary hormone deficiency with unknown etiology, and lays the foundation for investigating the role of EMT in pituitary tumor formation.

## Materials and methods

### Mice

Ames dwarf mice (*Prop1*$^{df/df}$) were originally obtained from Dr. A. Bartke (Southern Illinois University, Carbondale, IL) from a non-inbred stock (DF/B). Snell dwarf mice (*Pou1f1*$^{dw/dw}$) were originally obtained from The Jackson Laboratory from an inbred stock (DW/J) (Bar Harbor, ME) and crossed with *Mus castaneus*, CAST/EiJ mice for unrelated genetic mapping experiments. (DW/J-*Pit1*$^{dw}$ × CAST/EiJ) F$_2$ mice were used in this work. *Prop1*$^{-/-}$ mice (officially *Prop1*$^{tm1Sac}$) were generated from mouse embryonic stem cells engineered to contain a deletion in *Prop1* first exon through a portion of exon 2 (*Nasonkin et al., 2004*). All transgenic lines have been maintained as colonies at the University of Michigan through heterozygous matings. The morning after conception is designated e0.5, and the day of birth is designated as P1. All mice were housed in a 12-h light, 12-h dark cycle in ventilated cages with unlimited access to tap water and Purina 5058 chow. All procedures using mice were approved by the University of Michigan Committee on Use and Care of Animals, and all experiments were conducted in accordance with the principles and procedures outlined in the National Institutes of Health Guidelines of the Care and Use of Experimental Animals. The genotypes of *Prop1*$^{df/df}$ and *Pou1f1*$^{dw/dw}$ mice were determined as described (*Ward et al., 2005*; *Gage et al., 1996*). For all experiments, mutants were compared to wild type littermates, to control for the different genetic backgrounds.

### Generation of *Prop1-Tag* transgenic mice

A 25 kb transgene construct containing the mouse *Prop1* gene was used to generate *Prop1-Tag* transgenic mice (*Ward et al., 2007*). DNA encoding a 24 amino acid recognition motif for the *E. coli* biotinylation enzyme BirA was inserted at the ATG site in exon1, placing the Tag at the amino terminus of the PROP1 protein. The *Prop1-Tag* transgene was microinjected into the pronuclei of eggs from a C57BL/6J crossed with B6D2F1 mice, and the fertilized eggs were transferred into pseudo-pregnant foster mothers. Transgene positive progeny were detected by genotyping using PCR amplification of genomic DNA with primers that flank the junction of the *Tag* and *Prop1* coding sequence: 5'-GCTAGCCCCATGGCCTCTTCCCTGAGACA-3' and 5'- GCTAGCCATGGAAGAGCC TCCTGCGTT-3', under the following conditions: 92°C for 2 min, followed by 30 cycles of 92°C for 10 s, 57°C for 30 s and 72°C for 30 s, and a final 10 min extension at 72°C. Tg(Prop1-Tag)$^{SAC}$ were crossed with mice that ubiquitously express the BirA enzyme (FVB;129P2-Gt(ROSA)26Sor$^{tm1.1(birA)}$ $^{Mejr}$/J) (*Driegen et al., 2005*) to obtain *Prop1*$^{Tag}$; *Rosa26*$^{BirABirA}$ mice. For the rescue experiments, *Prop1*$^{Tag}$; *Rosa26*$^{BirABirA}$ mice were crossed with mice carrying the *Prop1* null allele, and *Prop1-Tag*; *Rosa26*$^{BirABirA}$; *Prop1*$^{+/-}$ progeny were crossed again with *Prop1*$^{+/-}$ to produce *Prop1-Tag*; *Rosa26*$^{BirABirA}$; *Prop1*$^{-/-}$mice (*Nasonkin et al., 2004*).

### Colony-forming assays

Cell culture assays for stem cell colony forming activity were carried out as previously described (*Gaston-Massuet et al., 2011*). Anterior pituitaries from *Prop1*$^{+/+}$, *Prop1*$^{df/df}$, *Pit1*$^{+/+}$, and *Pou1f1*$^{dw/dw}$ mice were collected and dissociated into a single-cell suspension following incubation in enzyme mix (0.5% w/v Collagenase type 2 (Lorne Laboratories), 0.1X Trypsin (Invitrogen), 50 µg/ml DNase I (Worthington) and 2.5 µg/ml Fungizone (Invitrogen) in Hank's Balanced Salt Solution (HBSS) (Gibco) for 4 hr at 37°C. Following washes in HBSS, cells were suspended in stem cell medium UltraCulture (LONZA) containing 5% Fetal Bovine Serum (Gibco), 20 ng/ml bFGF (R&D Systems) and 50 ng/ml cholera toxin (Sigma), counted and plated in six-well plates at a density of 8000 cells per well. After 10 days in culture, the cells were fixed with 4% formaldehyde for 20 min at room temperature, stained with Meyer's Hematoxylin and Eosin (SIGMA) or Cresyl Violet (SIGMA) for 2 min, and the colonies were counted manually. To calculate the number of cells within an entire gland that can form

colonies, the colony count was multiplied by the total number of cells recovered from the pituitary gland and the result divided by the number of cells plated (8000).

## RNA extraction and RT-qPCR

For RNA-Seq experiments and qRT-PCR, total RNA was isolated from cultured cells and pituitaries, using the RNAeasy Kit (Qiagen) following manufacturer's protocol. Total RNA was utilized to generate cDNA with Oligo dT Primers (Invitrogen) and Superscript II Reverse Transcriptase (Invitrogen) according to the manufacturer's instructions. The cDNA was used as a template in standard PCR reactions using TaqMan Universal PCR Master Mix (Applied Biosystems) according to the manufacturer's instructions. Reactions were run using an Applied Biosystems 7500 Real-Time PCR System. Expression of the housekeeping gene GAPDH served as an internal control for each RNA sample. The fold change values and standard deviations were calculated as described (*Vesper et al., 2006*; *Livak and Schmittgen, 2001*). For qPCR experiments from developmental pituitaries of dwarf and control mice, total RNA was extracted and the total was used to synthesize cDNA.

## RNA-Seq

After isolation of RNA, the Encore Complete RNA-Seq Library Systems (NuGen) protocol was followed using 500 ng of RNA per sample as starting material. Half of the ligated reaction volume was used for PCR (14 cycles) and the other half was kept at $-80°C$ as a backup. Libraries were checked for quality and quantified using the Bioanalyzer 2200 (Agilent, Santa Clara, CA), before being sequenced in barcoded pools on the Illumina Hiseq 2000 instrument (50 single sequencing, 3 lanes; Sequencing Core, UMICH).

## RNA-Seq analysis

Reads were aligned through Tophat version 2.0.8 using Bowtie version 2.1.0. Mouse Genome mm10 was used as the reference genome and the UCSC mm10.gtf was used for annotation and junction definitions. All default settings were used with Tophat except that a coverage-based search for junctions was disabled and all reads were mapped at all steps using the read-realign-edit-dist option. Differential expression was assessed using cufflinks version 2.1.1. and the cuffdiff command. Defaults were used for cuffdiff as well except for the multi-read-correct option which helps accurately weight reads that map to multiple locations. Data are output as FPKM values as defined by cufflinks. We extracted a set of significantly differentially expressed genes between the tested conditions (adj. p value $\leq 0.05$, |log2FC| $\geq 1$). For Pathways analysis, we imported Entrez Gene IDs into Ingenuity Pathways Analysis (IPA) (Ingenuity Systems; http://www.ingenuity.com). IPA calculates the score p-value that reflects the statistical significance of association between the genes and the pathways or networks by the Fisher's exact test. RNA-Seq data are available in the GEO database under accession number GSE77357.

## Immunostaining and in situ hybridization

Embryos and heads from P7 and P14 neonates were fixed for 2–4 hr in 4% formaldehyde in PBS (pH 7.2) at room temperature, dehydrated in a graded series of ethanol, and embedded in paraffin. Six-micrometer-thick sections were prepared for immunohistochemistry. CFCs were grown 10 days in culture on coverslips and fixed in 4% paraformaldehyde for 20 min at room temperature. Paraffin sections were incubated in sodium citrate buffer 10 mM, pH=8.5 at 80°C for 10 min to retrieve the antigens. Incubation with primary antibodies was performed overnight at 4°C. Sections and cells were washed twice in PBS for 20 min at room temperature and incubated 1 hr at room temperature with biotinylated secondary antibodies. A list of antibodies used in this work is provided in supplemental information. After washing twice in PBS, sections and cells were incubated with the Tyramide Signal Amplification (TSA) fluorescein isothiocyanate (FITC) kit (according to the manufacturer's protocol, Perkin-Elmer, Boston, MA). All sections and cells were incubated with DAPI to stain cell nuclei. Finally, they were mounted in ProlonGold (Life Technologies). All images were taken using Olympus FluoView 500 Laser Scanning Confocal Microscope. Embryonic pituitary paraffin sections were also prepared for in situ hybridization. An 809 bp fragment of the mouse *Zeb2* cDNA was cloned into pGEM-T easy (Promega) using the following primers: Fw 5' CATGCGAACTGCCATCTG3' and Rv 5'TATGCCTCTCGAGCTGGG 3'. The final vector was linearized and labeled with T7 polymerase.

The probe was diluted 1:100 and hybridized at 62°C. The riboprobe was generated and labeled with digoxigenin and precipitated with nitroblue tetrazolium chloride/5-bromo-4-chloro-3-indolyl phosphate (Roche Molecular Biochemicals, Indianapolis, IN) following previously described methods (*Suh et al., 2002*; *Cushman et al., 2001*).

## RNA interference methodology

Anterior pituitaries from *Prop1*$^{+/df}$ and *Prop1*$^{df/df}$ were collected at postnatal day 12 and dissociated into a single-cell suspension as described above. Following washes in HBSS, cells were suspended in stem cell medium UltraCulture (LONZA) containing 5% Fetal Bovine Serum (Gibco), 20 ng/ml bFGF (R&D Systems) and 50 ng/ml cholera toxin (Sigma), counted and plated in 12-well plates (353043, Falcon) at a density of 5000 cells per well. After 4 days in culture, colonies were treated with either 20 nM *Zeb2* Stealth siRNA (MSS216410, ThermoFisher Scientific), 20 nM negative control siRNA (AM4611, ThermoFisher Scientific) or left untreated. Lipofectamine RNAiMax (13778030, Thermo-Fisher Scientific) transfection reagent and the siRNAs were diluted independently in Opti-MEM tissue culture media (31985062, ThermoFisher Scientific). The diluted reagents were then combined and incubated at room temperature for 5 min. After the incubation, the siRNA and lipofectamine were added to the pituitary colonies. The colonies were incubated with or without the siRNA overnight after which the media was removed and replaced with fresh stem cell media. After 4 more days in culture the siRNA treatment was repeated. The colonies were grown for an additional 3 days then processed for cresyl violet staining or qRT-PCR. The ThermoFisher Taqman assays utilized were *Zeb2* (Mm00497193_m1), *Cdh1* (Mm0124357_m1) and *Gli2* (Mm01293117_m1).

## GHFT1 stable cell lines

GHFT1 cells (Dr Pamela Mellon, University of San Diego, La Jolla, CA) were maintained at 37°C/5% $CO_2$ in Dulbecco's Modified Eagle Medium (Invitrogen, Carlsbad, CA) supplemented with 10% heat inactivated fetal bovine serum (Hyclone, Logan, UT) and 100 units/ml penicillin-streptomycin (Invitrogen). All cultured cell lines employed in this study were authenticated by RNA-Seq analysis performed in the lab (*Figure 8—source data 1*). This authentication was done based on the positive expression of *Pou1f1* and absence of expression of *Prolactin, Growth hormome* and *Thyroid-Stimulating Hormone*. *Prop1* cDNA was cloned into pEF1α-FLBIO-puro (*Kim et al., 2009*) to be N--terminally tagged by standard cloning using the BamHI restriction site. The BirA expression vector used was pEF1α-BirAV5-neo (*Kim et al., 2009*). Both plasmids were a gift from Alan B. Cantor (Boston Children's Hospital, Boston, MA). Cells were plated onto 24-well plastic plates (Fisher Scientific, Fair Lawn, NJ) at a density of 3.5 × 10$^4$ cells/well. *Prop1* and BirA expressing vectors were transfected into cultured cells using the FuGENE 6 (Promega) at a 3:2 ratio according to the manufacturer's protocol. To make stable cell lines, transfected cells were incubated with 1.4 mg/ml G-418 solution (Roche) and 5 μg/ml Puromycin (Sigma) for selection of BirA and *Prop1-Tag,* respectively.

## Western blot analysis

Total protein from cultured cells was extracted in a T-PERbuffer (Thermo) and Halt protease and Phosphatase Inhibitor Cocktail (Thermo). Total protein was quantified using a BCA protein assay kit (Thermo). Proteins and ladder (BIO-RAD) were run on 4–20% Mini-ProteanTGX Gels from BIO-RAD. Membranes were washed with buffered Tris. Anti-V5 conjugated to horseradish peroxidase antibody (diluted 1:4000, Invitrogen) was used to detect BirA. Anti-Histone H3 antibody (Abcam) was used at a 1:4000 dilution and an anti-rabbit conjugated to horseradish peroxidase was used as a secondary antibody (diluted 1:20000, Thermo). All membranes were blocked and antibodies diluted in a 1% BSA/TBS-Tween solution. Membranes were developed with the SuperSignal West Pico chemiluminescent substrate (Thermo) and stripped with Restore western blot stripping buffer (Thermo).

## Chromatin immunoprecipitation

ChIP was performed as described in *Kim et al. (2009)* . Briefly, for ChIP-Seq, approximately 10$^7$ cells were cross-linked with 1% formaldehyde at room temperature for 10 min and neutralized with 0.15 M glycine. After sonication, 10 μg of soluble chromatin was incubated with streptavidin magnetic beads (Dynabeads MyOne Streptavidin T1, Invitrogen) at 4°C overnight. Subsequently, beads were washed and DNA was eluted with elution buffer 1% SDS. After reverse crosslinking, RnaseA

and proteinase K digestion, chromatin was purified using a Qiagen PCR purification kit. For ChIP-Seq, the extracted DNA was ligated to specific adaptors followed by deep sequencing with the Illumina HiSeq 2000 system according to the manufacturer's instructions. The confirmatory ChIPs for specific peaks were performed with immunoprecipitated and reverse cross-linked chromatin that was prepared as described above. Quantitative real time PCR was performed with primers specific for ChIP-Seq peak sequences using SYBR Green (Applied Biosystems) assays. Values were normalized to input. The primers used for real-time PCR are listed on Supplemental Information.

## ChIP-Seq

Multiplex libraries were prepared using Rubicon ThruPlex kit according to their protocol (Cat R40012). Library amplification reaction was performed in 14 cycles. The concentration and quality of purified, amplified DNA were estimated using a high sensitivity DNA assay Bioanalyzer 2200 (Agilent). After quality confirmation, the DNA libraries were sequenced on an Illumina High Seq 2000 (Sequencing Core, UMICH).

## Chip-Seq analysis and motif enrichment

DNA sequence reads were mapped to the reference genome mm10 assembly by Bowtiev0.12.7. MACS2 was used to identify peaks. FastQC and cross-correlation test were performed to evaluate the data quality. Peaks were annotated for their locations regarding to nearby genes by HOMERv4.6. To discover DNA binding motifs enriched among PROP1 peaks, HOMER's findMotifsGenome.pl was used. FIMO was used to identify locations of motifs inside peaks. ChIP-Seq data are available in the GEO database under accession number GSE77302.

## Acknowledgements

Funding was provided by the National Institutes of Health (R01HD-30428 to SAC), the Endocrine Society International Scholar Program and University of Michigan Center for Organogenesis Fellowship (MIPM). We thank Thom Saunders, Wanda Filipiak and Maggie Van Keuren of the University of Michigan Transgenic Animal Model Core, which was supported by NIH grants CA46592, AR20557, and DK34933. We thank Robert Lyons and Craig Johnson of the University of Michigan Sequencing Core and Weisheng Wu from the University of Michigan Bioinformatics Core. We thank Dr. Pamela Mellon from University of California San Diego for the GHFT1 cells and Dr. Alan Cantor, Children's Hospital Boston, for the Biotin Tagging system vectors. We thank Dr. Aimee Ryan, Montréal, Quebec for the guinea pig anti-PROP1 antibody. We thank Dr. Luciani Carvalho and Claudia Chang of U São Paulo, Brazil for their contributions to the early stages of the study. We thank Bryan Hahm for technical assistance.

## Additional information

### Funding

| Funder | Grant reference number | Author |
|--------|------------------------|--------|
| National Institutes of Health | R01HD-30428 | Sally A Camper |
| University of Michigan | Fellowship. Center of Organogenesis | María Inés Pérez Millán |
| Endocrine Society | Endocrine Society International Scholar Program | María Inés Pérez Millán |

The funders had no role in study design, data collection and interpretation, or the decision to submit the work for publication.

### Author contributions

MIPM, Conception and design, Acquisition of data, Analysis and interpretation of data, Drafting or revising the article; MLB, AHM, Acquisition of data, Analysis and interpretation of data, Drafting or

revising the article; SAC, Conception and design, Analysis and interpretation of data, Drafting or revising the article

### Author ORCIDs
Sally A Camper, http://orcid.org/0000-0001-8556-3379

### Ethics
Animal experimentation: This study was performed in strict accordance with the recommendations in the Guide for the Care and Use of Laboratory Animals of the National Institutes of Health. All of the animals were handled according to approved institutional animal care and use committee (IACUC) protocol of the University of Michigan. The protocol was approved by the University Committee on Use and Care of Animals (UCUCA) of the University of Michigan (PRO00004640).

## Additional files

### Supplementary files
• Supplementary file 1. Supplementary Information. (A) Primary antibodies used in this study. (B) Secondary Antibodies in this study. (C) Primers for RT-PCR used to confirm ChIP-Seq peaks.

### Major datasets
The following datasets were generated:

| Author(s) | Year | Dataset title | Dataset URL | Database, license, and accessibility information |
|---|---|---|---|---|
| Perez Millan MI, Brinkmeier M, Camper S | 2016 | PROP1 triggers epithelial-mesenchymal transition-like process in pituitary stem cells (CHIP-Seq) | http://www.ncbi.nlm.nih.gov/geo/query/acc.cgi?acc=GSE77302 | Publicly available at the NCBI Gene Expression Omnibus (Accession no: GSE77302). |
| Perez Millan MI, Brinkmeier M, Camper S | 2016 | PROP1 triggers epithelial-mesenchymal transition-like process in pituitary stem cells (RNA-Seq) | http://www.ncbi.nlm.nih.gov/geo/query/acc.cgi?acc=GSE77357 | Publicly available at the NCBI Gene Expression Omnibus (Accession no: GSE77357). |

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
