## [Decision Letter]

Thank you for submitting your article "PROP1 triggers epithelial-mesenchymal transition-like process in pituitary stem cells" for consideration by *eLife*. Your article has been reviewed by three peer reviewers, and the evaluation has been overseen by Janet Rossant as the Senior Editor. The following individual involved in review of your submission has agreed to reveal their identity: JP Martinez-Barbera.

The reviewers have discussed the reviews with one another and the Reviewing Editor has drafted this decision to help you prepare a revised submission.

Summary:

In an interesting study, Pérez Millan et al. provide molecular evidence supporting the idea that the PROP1 transcription factor initiates an EMT-like process during pituitary development. This idea was implicit in descriptions of PROP1 mutant mice initiated almost 20 years ago and pursued by Dr Camper, but the present work provides RNA-seq and ChIP-seq data for PROP1 in the support of this hypothesis. Changes in gene expression in PROP1 mutant pituitaries, such as for the genes encoding E-cadherin, claudin 23, MP2/3, ZEB2, *Notch2* and GLI2, are correlated with presence of PROP1 ChIP-seq peaks in their vicinity. The authors also show that PROP1 expression overlaps with the progenitor marker *Sox2*, presumably as progenitors exit the stem state towards differentiation. They also show that the transitional marker cyclin E is much less expressed in PROP1 mutant pituitaries despite the fact that the cell cycle inhibitors p57 and p27 are expressed normally in developing pituitaries. Finally, they show that the colony-forming ability of PROP1 mutant pituitaries is qualitatively, but not quantitatively, altered to yield more of a differentiated rather than progenitor appearance.

While the data presented are interesting, provocative and highly suggestive of the importance of a PROP1-dependent EMT-like process, the reviewers were concerned that the evidence presented is largely correlative. As detailed in the individual critiques below, the major common concern was that the data on changes in E-cadherin expression were not very convincing and that some form of more functional assessment of the proposed target genes in PROP1 mutants would help support the hypothesis proposed in the title of the paper. While there are a number of issues raised by the referees, the three key areas that must be addressed before we can consider a revised manuscript are listed below. If you feel you can address these in the required two-month time frame, we will be happy to receive a revised manuscript. If you have concerns about this, please let us know and we can discuss further. It may help us if you responded to this decision letter with a list of the experiments you feel address the major concerns of the reviewers and a reasonable timetable for their completion.

Essential revisions:

1) The role of *Prop1* in EMT is not characterized very well. Although potential target genes involved in EMT are identified by ChIP-seq, the E-Cadherin immunostaining is not convincing. Functional characterization of the importance of some of the target genes would help support the conclusions. At least one of the proposed PROP1-dependent regulators of EMT should be assessed by mutagenesis in order to offer a convincing piece of work. Perhaps the authors could try a functional approach in the adherent culture system.

2) A better characterization of the *Sox2* and *Prop1*-expressing cells postnatally is required.

3) It is unexpected to see formation of the same number of colonies in vitro, when in vivo the proportion of *Sox2* positive cells is increased in *Prop1* mutants. The authors comment that mutant colonies do not have the same morphology as wild-type ones so maybe prevention of the EMT like-transition is perturbing colony formation? Clonogenic differences need to be better presented and discrepancies with *Sox2*-immunostaining need to be clarified.

---

## [Author Response]

Essential revisions:

1) The role of Prop1 in EMT is not characterized very well. Although potential target genes involved in EMT are identified by ChIP-seq, the E-Cadherin immunostaining is not convincing. Functional characterization of the importance of some of the target genes would help support the conclusions. At least one of the proposed PROP1-dependent regulators of EMT should be assessed by mutagenesis in order to offer a convincing piece of work. Perhaps the authors could try a functional approach in the adherent culture system.

We have replaced the E-cadherin staining in Figure 5 panel B, which shows abnormal, persistent expression in colonies from *Prop1* mutants. We also show a merged image of E-cadherin and N-cadherin in Figure 7 Panel A that makes it easier to appreciate the excess E-cadherin staining in the *Prop1* mutant pituitary during embryonic development and a new Panel B that shows ectopic E-cadherin staining in *Prop1* mutant pituitaries postnatally.

We reasoned that the functional characterization of a down-stream target of *Prop1* by knock out or overexpression would be more likely to generate an obvious phenotype if the target gene selected is itself a master-regulator like *Zeb2* instead of a member of one of the larger gene families that were implicated like cadherins, matrix-metalloproteinases, or BMPs. We used si-RNA to knock down *Zeb2* expression in the adherent pituitary stem cell colonies derived from normal mice and found that this was sufficient to cause reduced *Gli2* expression, increased *Cdh1* expression and alter colony morphology such that they resemble colonies derived from *Prop1* mutant mice. This appears in the new Figure 6.

2) A better characterization of the Sox2 and Prop1-expressing cells postnatally is required.

We created a new Figure 3 that shows immunohistochemical staining for SOX2, SOX9 and PROP1 in normal and *Prop1* mutant mice in postnatal pituitary development. We quantified the fraction of cells that are double positive, and the data reveal reduced co-expression of *Sox2* and *Sox9* in *Prop1* mutants. This is consistent with the idea that the SOX2 positive cells in *Prop1* mutants are failing to engage in the transition from stem cell towards differentiation.

*3) It is unexpected to see formation of the same number of colonies in vitro, when in vivo the proportion of Sox2 positive cells is increased in Prop1 mutants. The authors comment that mutant colonies do not have the same morphology as wild-type ones so maybe prevention of the EMT like-transition is perturbing colony formation? Clonogenic differences need to be better presented and discrepancies with Sox2-immunostaining need to be clarified*.

We changed the presentation of the colony forming properties as suggested by the reviewers. Instead of showing the percentage of the cells that can form colonies, we present the total number of colony forming cells per organ. Because the *Prop1* and *Pit1* mutant pituitaries lack the cells that make GH, PRL and TSH, the representation of CFC/organ is probably more appropriate. See Figure 4, Panel A which illustrates significantly fewer CFC/*Prop1* mutant pituitary and no change in CFC/*Pit1* mutant pituitary. Although there are more *Sox2* expressing cells in *Prop1* mutant pituitaries, fewer have clonogenic potential, likely because of the reduced number of SOX2, SOX9double positive cells as described above for point 2.